# Neighborhood satisfaction and reproductive status

Zachary P. Neal *, Jennifer Watling Neal

Psychology Department, Michigan State University, East Lansing, MI, United States of America

* zpneal@msu.edu

## Abstract

Both urban planners and urban scholars have been keenly interested in identifying the characteristics associated with neighborhood satisfaction. One robust but surprising pattern is that the presence or number of children in a household has no effect on neighborhood satisfaction. To clarify this pattern, we measured the neighborhood satisfaction of a representative sample of 1,000 Michigan adults, whom we divided into six distinct reproductive statuses: co-parents, single-parents, empty nesters, not-yet-parents, childless individuals, and childfree individuals. We found that a simple parent vs. non-parent dichotomy hides significant heterogeneity among these groups. Specifically, we found that single parents and childfree individuals experience significantly less neighborhood satisfaction than other groups. We conclude by reflecting on the methodological and practical implications of differences in neighborhood satisfaction when more nuanced reproductive statuses are considered.

**Data Availability Statement:** All relevant data can be found: https://osf.io/k2w5c.

**Funding:** The author(s) received no specific funding for this work.

## 1 Introduction

Both urban planners and urban scholars have been keenly interested in identifying the characteristics associated with neighborhood satisfaction, with the long-term goal of enhancing the well-being and happiness of all segments of the population [1]. However, one particularly robust but puzzling pattern is that the presence or number of children in a household has no statistically significant association with the respondent's neighborhood satisfaction [2–19]. In this paper, we explore this surprising finding by asking: Does such an important life choice as having children or being a parent really not matter for neighborhood satisfaction?

We contend that prior studies comparing households with and without children, or comparing parents to non-parents, fail to capture important differences that exist across different reproductive statuses. In this study, we examine unique data from a representative sample of 1,000 Michigan adults that allows us to distinguish six different reproductive statuses: Co-parents, single parents, empty nesters, not-yet-parents, childless individuals, and childfree individuals. By considering these groups separately, we find that single parents and childfree individuals are significantly less satisfied with their neighborhoods than individuals with other reproductive statuses. These findings are significant for the study of neighborhood satisfaction because they suggest that simple classifications based on the number or presence of children

**Competing interests:** The authors have declared that no competing interests exist.

obscure important differences. They are also significant for policies aimed at building satisfying neighborhoods because they suggest that neighborhoods are less satisfying for a large segment of the population.

The remainder of the paper is organized in four sections. In the background section, we briefly review the neighborhood satisfaction literature, focusing on neighborhood satisfaction's association with reproductive characteristics such as parental status and the presence of children. In the methods section, we describe the Michigan State of the State Survey (SOSS), including the measurement strategy used to differentiate six reproductive statuses. In the results section, we report two regression models and a series of mean comparisons focusing on differences in satisfaction across these six groups. In the discussion section, we conclude by reviewing the study's limitations, identifying directions for future research, and highlighting the findings' implications for research and policy.

## 2 Background

### 2.1 Neighborhood satisfaction

On the surface, the construct of "neighborhood satisfaction" seems straightforward: it describes the extent to which residents are satisfied with the neighborhoods in which they live. However, in practice, studies of neighborhood satisfaction rarely go beyond this intuitive or commonsense definition. Neighborhood satisfaction can be understood as an assessment of "an individual's attitude (satisfaction) toward an object (a neighborhood)" [20], which means that defining neighborhood satisfaction requires defining both 'a neighborhood' and 'satisfaction.' However, here too there is significant ambiguity in the literature. The boundaries of the reference neighborhood are often left up to individual respondents [21, 22], while in other cases they are defined by census tracts or other formal geographies [8, 23, 24]. Similarly, satisfaction (in the context of satisfaction with neighborhoods) has been defined variously as a unidimensional attitude [21], a multidimensional attitude composed of one's satisfaction toward specific aspects of the neighborhood [25], or as one dimension of a multidimensional construct of global life satisfaction [26].

Despite the lack of clarity on what neighborhood satisfaction is, there are several well-developed theoretical perspectives on where neighborhood satisfaction comes from. First, the *urban scale* [27] or *ecological* [21] perspective contends that neighborhood satisfaction is driven by objective features of the neighborhood, such as the availability of specific amenities (e.g., parks) or the absence of undesirable features (e.g., crime). Second, the *compositional* [27] or *systemic* [21] perspective contends that neighborhood satisfaction is driven by individuals' demographic characteristics. Finally, the *subjective* perspective [27] contends that "an individual's satisfaction with any set of circumstances is dependent. . .on a whole set of values, attitudes, and expectations" that can lead to "discrepancies between reality and perception" [28], and thus that neighborhood satisfaction is driven by an individual's subjective perceptions of the neighborhood.

These theories of neighborhood satisfaction have set researchers on a path toward identifying the correlates of neighborhood satisfaction in three broad categories: objective features, demographic characteristics, and subjective perceptions [29]. Recent studies provide more support for the subjective perspective and role of perception, than for the ecological perspective and role of objective features. One large study of adults in the United States ($N$ = 1726) found that residents' perceptions of neighborhood characteristics such as land use and park access were significantly associated with their neighborhood satisfaction, while the neighborhood's actual land use and park access were not [25]. Similarly, a meta-analysis of 27 studies of neighborhood satisfaction, found wide variation in the neighborhood satisfaction of residents

in the same neighborhood, and thus limited evidence that the neighborhoods' objective characteristics are associated with neighborhood satisfaction [20].

Past studies also provide limited support for the compositional perspective and role of some demographic characteristics. For some demographic characteristics such as education, the association is unclear, with studies finding it is is positively [11], negatively [8], or not [30] associated with neighborhood satisfaction. However, for other demographic characteristics such as age and gender, the association is robust: women and older adults are more satisfied with their neighborhoods [8, 11, 30]. Findings concerning the role of children and parental status in neighborhood satisfaction have also been remarkably consistent, but in a surprising direction. Whether considering the *presence* of children [2–13, 18] or the *number* of children in the household [14–17, 19], studies consistently find that children and parental status are not associated with neighborhood satisfaction. To our knowledge, only one recent study found the presence of children to be significantly associated with neighborhood satisfaction, however the effect was weak ($\beta = 0.037$) [31].

There are two reasons it is surprising that studies have consistently found no association between parental status and neighborhood satisfaction. First, it is widely theorized that parenthood is associated with global well-being and life satisfaction, albeit with some disagreement about the direction of the association. Folk theories suggest that having children makes one's life more fulfilling [32, 33], while more recently others have recognized that the resource demands of parenthood may reduce happiness [34]. Second, some neighborhood satisfaction research adopts a perspective that implicitly values nuclear family parenthood. For example, one study interpreted the findings by speculating that "childless adults. . .generally lack the opportunities for interaction experienced by. . .residents with children" [35], while another suggested that "the combination of divorced households and households with children is a measure of broken households" [8]. This child-centric perspective can also impact researchers' measurement decisions, for example when neighborhood satisfaction is measured by asking "How would you rate your neighborhood *as a place to raise children*," (emphasis added) [16]. Ultimately, this perspective leads some authors to recommend that neighborhood-building should include "investments in policies and activities that make a community. . .a better place for families" [36].

## 2.2 Reproductive status

Prior studies of neighborhood satisfaction have examined associations with the presence or number of children, but these demographic characteristics fail to capture differences in individuals' reproductive statuses and life-stages. By focusing on whether or how many children are present, these approaches implicitly split respondents into two groups. The group of respondents whose households have children includes (1) *single parents* with children at home, (2) partnered or *co-parents* with children at home, and (3) non-parents who live in a household with children (e.g., adult siblings, aunts/uncles). However, these types of respondents are potentially quite different, and may have different experiences of neighborhood satisfaction. The group of respondents whose households do not have children includes (1) *empty nesters* whose children have moved out, (2) *not-yet-parents* who do not have children yet but plan to have them, (3) *childless* people who wanted children but could not have them, and (4) *childfree* people who do not want children. Again, these types of respondents are potentially quite different, and may have different experiences of neighborhood satisfaction. Thus, the categorization of respondents by the presence of children ignores potentially important differences in reproductive decisions and life-stages.

Although most studies of neighborhood satisfaction have not examined differences in reproductive statuses, there have been some exceptions. Some studies have examined heterogeneity in partnership status among parents, yielding mixed results. One study found no significant differences in neighborhood satisfaction between co-parents and single parents [37], while another found that single parents were less satisfied with their neighborhood than co-parents [38]. An additional study examined the role of heterogeneity in life-stage in individuals' perceptions of neighborhood family friendliness, finding that couples with young children had more favorable perceptions of neighborhood family friendliness than people with no children, parents with adolescents or adult children living at home, and empty nesters [39].

Despite currently limited and inconclusive research, there is reason to believe that reproductive statuses might matter for neighborhood satisfaction. Personal characteristics like reproductive statuses may lead to different residential needs that affect neighborhood satisfaction, which is maximized when the resident and residential environment are well-matched [20, 40, 41]. For example, single parents and co-parents may seek out neighborhood amenities that are child-friendly such as safety while childfree individuals may be interested in neighborhood amenities such as nightlife [11]. The concept of *life stage–neighborhood fit*, defined as "the degree to which a neighborhood is a good fit for the interests, abilities, and needs of families at varying life stages" [39], builds on this idea and broader theories of person-environment fit [41]. In addition to neighborhood amenities, it emphasizes that neighborhood satisfaction may also be driven by a demographic match between an individuals' own reproductive status and that of their surrounding neighbors. For example, in their study, co-parents were more likely to perceive their neighborhood as "family friendly" when they lived in a neighborhood with many other families with children.

Prior studies of neighborhood satisfaction have failed to find an association with the presence or number of children in a household. One possibility is that such an association has been obscured by hidden heterogeneity among respondents in households with children, and among respondents in households without children. In this study, we investigate that possibility by asking two related questions. First, we ask *is reproductive status associated with neighborhood satisfaction?* (**Research Question 1**). Second, we ask *how satisfied is each reproductive status group with their neighborhood, and which groups differ in their neighborhood satisfaction* (**Research Question 2**).

## 3 Methods

### 3.1 Data

This study uses State of the State Survey (SOSS) data collected by the Michigan State University Institute for Public Policy and Social Research (IPPSR) between May 8th and 25th, 2020. Data was collected from 1,000 respondents, who were then matched to a sampling frame constructed from the 2016 1-year American Community Survey on sex, age, race, and education to compute survey weights. Respondents were recruited throughout the state of Michigan, including from rural and urban areas, with counties represented roughly proportionally to their population size. Data collection occurred during the initial wave of the COVID-19 pandemic when residents were under a stay-at-home order from the Governor. Data collection ended on the day that George Floyd was murdered by a police officer in Minneapolis, resulting in widespread protests against police brutality and systemic racism. Based on the survey's timing, respondents' reported neighborhood satisfaction may have been affected by the COVID-19 pandemic, but was not influenced by the murder of George Floyd or subsequent protests [42, 43].

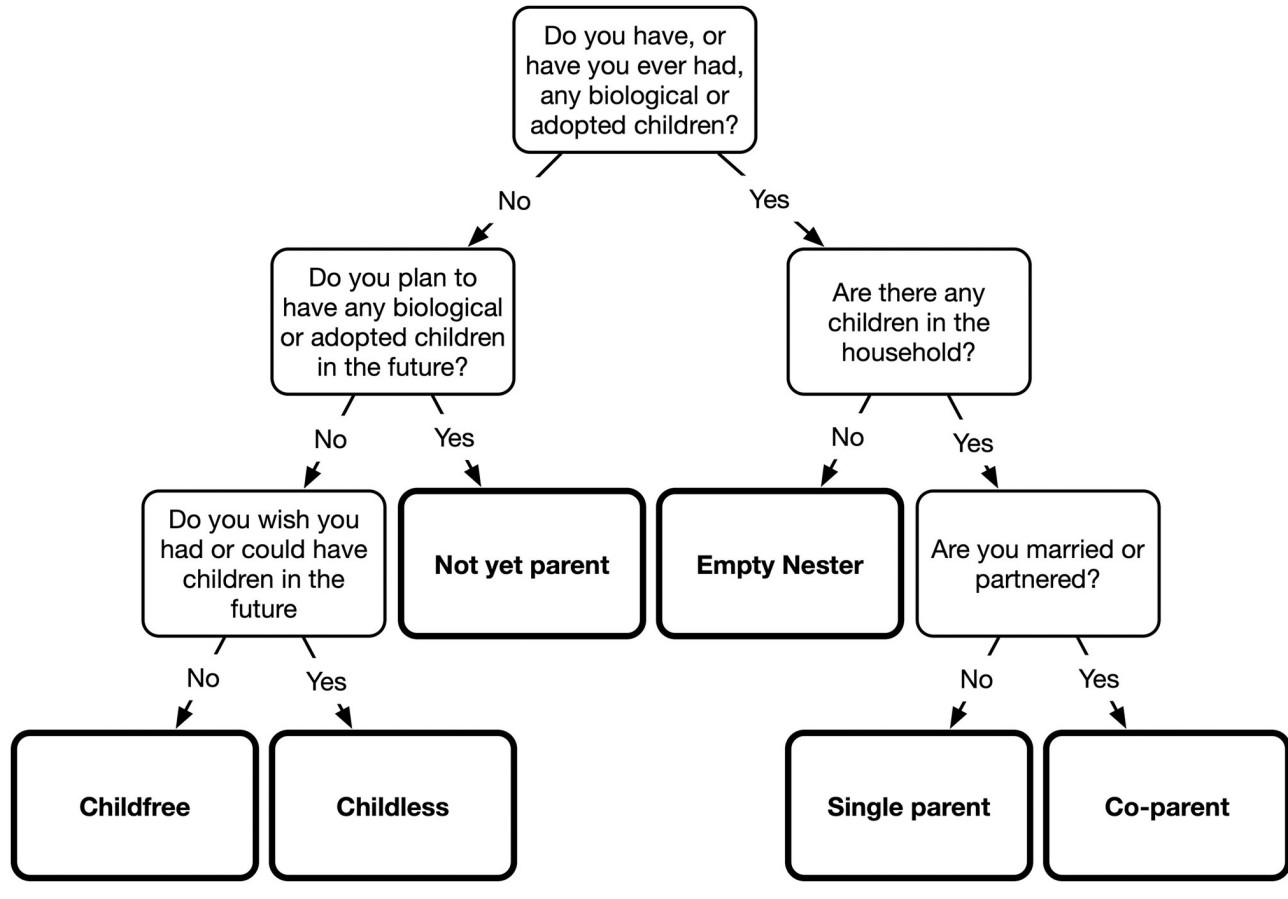

**Fig 1. Decision tree for classifying respondents by reproductive status.**

### 3.2 Measures

**3.2.1 Reproductive status.** We classified each respondent into six mutually exclusive reproductive statuses by asking up to three binary questions from a set of five using the decision tree illustrated in Fig 1. A respondent who has had children (i.e., a parent) is classified as an *empty nester* if there are no children in the household, as a *co-parent* if they are married or partnered and there are children in the household, or as a *single parent* if they are not married or partnered and there are children in the household. A respondent who has not had children (i.e., a non-parent) is classified as a *not-yet-parent* if they plan to have children in the future, as *childless* if they want(ed) children but could not have them, or as *childfree* if they do not want children. This mirrors the approach used in earlier studies to distinguish different types of non-parents [44, 45], but also distinguishes different types of parents. In all models, co-parents are the omitted or reference category.

**3.2.2 Neighborhood satisfaction.** We measure neighborhood satisfaction by asking: *Taking everything into account, how satisfied are you with your neighborhood as a place to live?* Responses were provided on a 5-point ordinal scale ranging from 'very satisfied' (5) to 'very dissatisfied' (1). This approach follows the widely-used convention of measuring neighborhood satisfaction with a single-item [3, 11, 16, 29, 46–49]. This item leaves the definition of 'neighborhood' up to each respondent, which can be problematic if the goal is to draw conclusions about specific geographically delimited areas (e.g., the effect of green space in tract X on

satisfaction with tract X). However, this approach to measuring neighborhood satisfaction is well-suited to this study, which is focused on individuals' phenomenological experience of the place they perceive as 'their neighborhood' [22].

**3.2.3 Covariates.** Because our data come from a representative sample of the population, we estimate the association between reproductive status and neighborhood satisfaction without covariates to identify mean differences among the reproductive status groups. To rule out some possible confounding individual-level effects, we also estimate this association controlling for potentially influential individual characteristics. First, we control for the sociodemographic characteristics of sex, race, age, age$^2$, and education. Sex was coded as a binary variable where 1 reflects males and 0 reflects females. Due to limited variation in respondents' race and ethnicity, race was coded as a binary variable where 1 reflects respondents who identified as White alone and not Hispanic, while 0 reflects respondents who identified as Hispanic and/or as a Person of Color. Education was measured as the highest degree completed by respondents and was coded as a 7-point ordinal scale: 1 (Less than High School), 2 (High School Diploma or GED), 3 (Some College) 4 (Technical College/Junior College Graduate) 5 (4 Year College Graduate), 6 (Some Graduate School) and 7 (Graduate Degree). We calculated age by subtracting respondents' answers to the question, *In what year were you born?*, from 2020 (i.e., the year the survey data were collected).

Second, we control for respondents' subjective well-being, measured using the five-item satisfaction with life scale (SWLS) [50]. The SWLS includes five items: *In most ways my life is close to my ideal*; *The conditions of my life are excellent*; *I am satisfied with my life*; *So far I have gotten the important things I want in life*; and *If I could live my life over, I would change almost nothing.* Each item is rated on a 7-point scale ranging from 1 (*strongly disagree*) to 7 (*strongly agree*). We computed a scale score by averaging across the five items, which yielded a scale with high inter-item reliability ($\alpha = 0.906$). Satisfaction with life is a key dimension of subjective well-being, which is known to be associated with neighborhood satisfaction [29, 48, 49, 51]. Controlling for satisfaction with life allows us to isolate the association between respondents' reproductive status and neighborhood satisfaction that is independent of their subjective well-being, which may be driven by a range of unmeasured physical health, mental health, and environmental factors.

Third, because these data were collected during the COVID-19 pandemic, we control for respondents' COVID stress, which is likely to be negatively associated with neighborhood satisfaction. COVID stress is measured by asking: *How has the COVID-19 impacted how stressed or anxious you are overall?* Responses were provided on a 5-point ordinal scale ranging from 'much less stressed/anxious' (1) to 'much more stressed/anxious' (5).

## 3.3 Analytic plan

Prior to analysis, all continuous variables were grand-mean centered to aid interpretation. All analyses were conducted using the `survey` package for **R**, using survey weights computed to obtain estimates that are representative of Michigan's adult population [52]. A small number of respondents were missing data on one or more analytic variables. Because these data are missing completely at random (MCAR: $\chi^2_{44} = 51.54$, $p = 0.203$; [53]) and because multiple imputation is not possible with weighted survey data, cases with missing data were dropped listwise, yielding an analytic sample of 946. We use the conventional $p < 0.05$ threshold for detecting statistically significant effects. The data and code necessary to reproduce the analyses reported below is available at https://osf.io/k2w5c.

**Table 1. Population descriptives.**

| | Mean | SE | Pearson Correlation | | | | | | |
|---|---|---|---|---|---|---|---|---|---|
| Co-parent | 0.147 | 0.015 | | | | | | | |
| Not-Yet-Parent | 0.122 | 0.017 | | | | | | | |
| Childless | 0.075 | 0.01 | | | | | | | |
| Empty Nester | 0.324 | 0.018 | | | | | | | |
| Childfree | 0.269 | 0.019 | | | | | | | |
| Single Parent | 0.064 | 0.012 | | | | | | | |
| | | | Male | White | Education | Age | SWLS | Stress | |
| Male | 0.482 | 0.021 | — | | | | | | |
| White | 0.768 | 0.02 | 0.045 | — | | | | | |
| Education | 3.442 | 0.08 | 0.039 | 0.009 | — | | | | |
| Age | 49.791 | 0.845 | -0.037 | 0.156** | -0.001 | — | | | |
| SWLS | 4.217 | 0.067 | 0.007 | 0.088** | 0.116** | 0.161** | — | | |
| COVID Stress | 3.695 | 0.036 | -0.148** | 0.068* | 0.046 | -0.099** | -0.237** | — | |
| N'hood Satis. | 3.887 | 0.047 | -0.014 | 0.085* | 0.06 | 0.197** | 0.382** | -0.202** | |

N = 946, weighted sample;

$^*p < 0.05$,

$^{**}p < 0.01$

## 4 Results

### 4.1 Sample

Table 1 reports the descriptive characteristics of our weighted sample. We observe that the most prevalent reproductive status in Michigan is empty nesters (32.4%), followed closely by childfree individuals (26.9%), reflecting Michigan's aging population and prior findings about the large but previously hidden childfree population [44, 45]. Co-parent (14.7%) and not-yet-parent (12.2%) statuses have a similar prevalence, while childless individuals (7.5%) and single parents (6.4%) are rarer in the population. Because the sample is weighted to ensure representativeness by sex, age, race, and education, estimated means of these characteristics in our sample match those of the Michigan population. The bivariate correlations display expected associations, for example, that neighborhood satisfaction is positively associated with life satisfaction ($r = 0.382$, $p < 0.01$), but negatively associated with COVID stress ($r = −0.202$, $p < 0.01$).

Our study of reproductive status differences in neighborhood satisfaction was motivated by the consistent finding that children are not associated with neighborhood satisfaction. Before turning to models designed to answer our research questions, we first wanted to confirm that our sample also replicated this past finding. We find that at the bivariate level, the presence of children in the household ($b = −0.120$, $p = 0.265$) and the number of children in the household ($b = −0.09$, $p = 0.07$) are not statistically significantly associated with neighborhood satisfaction. Table 2 illustrates that this lack of a significant association persists even after controlling for demographic characteristics when considering either the presence ($b = 0.132$, $p = 0.260$) or number of children ($b = 0.001$, $p = 0.986$) in the household. These preliminary models, which replicate prior findings, suggest that our sample is similar to those investigated in other studies of parenthood and neighborhood satisfaction, and thus offers an ideal case for understanding how a more nuanced operationalization of reproductive status may explain this apparent lack of association.

**Table 2. Association of the presence or number of children and neighborhood satisfaction.**

|  | Model 1 | Model 2 |
|---|---|---|
| Intercept | 3.643 (0.143)** | 3.684 (0.139)** |
| Presence of children | 0.132 (0.118) | — |
| Number of children | — | 0.001 (0.056) |
| Male | -0.014 (0.09) | -0.018 (0.091) |
| White | 0.152 (0.127) | 0.15 (0.128) |
| Education | 0.049 (0.026) | 0.048 (0.026) |
| Age | 0.016 (0.003)** | 0.015 (0.003)** |
| Age$^2$ | 0 (0)** | 0 (0)** |
| R$^2$ | 0.064 | 0.061 |

N = 946 weighted sample;

$^*p < 0.05$,

$^{**}p < 0.01$

## 4.2 Research Question 1: Does reproductive status matter?

Model 1 in Table 3 tests whether reproductive status matters for neighborhood satisfaction, and specifically whether different reproductive status groups have different mean levels of neighborhood satisfaction. The $R^2$ indicates that reproductive status explains 6.7% of the variation in neighborhood satisfaction, while the significant F-change statistic ($F_{5,940} = 81.003$, $p < 0.01$) indicates that including reproductive status explains a significant amount of variation relative to a null model. Examining the estimated coefficients, we find that co-parents'

**Table 3. Model estimates (DV: Neighborhood Satisfaction).**

| Variable | Model 1 | Model 2 |
|---|---|---|
| Intercept | 4.125 (0.107)** | 4.076 (0.141)** |
| Not-Yet-Parent | -0.232 (0.144) | -0.063 (0.165) |
| Childless | -0.205 (0.178) | -0.123 (0.163) |
| Empty Nester | 0.011 (0.129) | -0.2 (0.138) |
| Childfree | -0.494 (0.15)** | -0.403 (0.133)** |
| Single Parent | -1.022 (0.22)** | -0.698 (0.238)** |
| Male | – | -0.094 (0.08) |
| White | – | 0.089 (0.109) |
| Education | – | 0.009 (0.023) |
| Age | – | 0.01 (0.003)** |
| Age$^2$ | – | 0 (0) |
| SWLS | – | 0.212 (0.03)** |
| COVID Stress | – | -0.143 (0.048)** |
| R$^2$ | 0.067 | 0.208 |
| ΔF | $F_{5,940} = 81.003^{**a}$ | $F_{5,933} = 30.637^{**b}$ |

N = 946 weighted sample;

$^*p < 0.05$,

$^{**}p < 0.01$

[a] Compared to a reduced model with no covariates

[b] Compared to a reduced model with only covariates

**Table 4. Mean neighborhood satisfaction.**

| Group | Mean[a] | Different from[b] |
|---|---|---|
| (A) Co-Parent | 4.076 | EF |
| (B) Not-Yet-Parent | 4.013 | EF |
| (C) Childless | 3.953 | F |
| (D) Empty Nester | 3.877 | F |
| (E) Childfree | 3.673 | AB |
| (F) Single Parent | 3.378 | ABCD |

N = 946 weighted sample

[a] Controlling for model 2 covariates.

[b] $p < 0.05$

mean neighborhood satisfaction (the intercept in these models) is 4.125 on a 5-point scale, while childfree individuals ($b = -0.494$, $p < 0.01$) and single parents ($b = -1.022$, $p < 0.01$) have significantly lower neighborhood satisfaction. Because neighborhood satisfaction is measured on a 5-point scale, these differences are large in absolute terms, with single parents reporting an average neighborhood satisfaction a full point lower than co-parents.

Model 2 in Table 3 repeats this test, but includes several covariates that may be associated with both reproductive status and neighborhood satisfaction. Including these covariates increases the model's overall explanatory power ($R^2 = 0.208$), but the significant F-change statistic ($F_{5,933} = 30.637$, $p < 0.01$) confirms that reproductive status explains a significant amount of variation beyond that explained by the covariates alone. When these covariates are included, we observe a similar pattern in the estimated coefficients: co-parents' mean neighborhood satisfaction is 4.076, while the mean neighborhood satisfaction of both childfree individuals ($b = -0.403$, $p < 0.01$) and single parents ($b = -0.698$, $p < 0.01$) is significantly lower. Therefore, both with and without covariates, we find that *reproductive status does matter for neighborhood satisfaction.*

## 4.3 Research Question 2: Which groups are satisfied?

To determine which reproductive status groups differ in their average neighborhood satisfaction, we first estimated each group's mean neighborhood satisfaction, controlling for the covariates included in Model 2. We then performed mean-different t-tests for each pair of groups. Table 4 shows each group's mean neighborhood satisfaction, and the pairs of groups with statistically significantly different means at the $\alpha = 0.05$ level. We find that co-parents are on average most satisfied with their neighborhood ($M = 4.076$), followed closely by not-yet-parents ($M = 4.013$), childless individuals ($M = 3.953$), and empty nesters ($M = 3.877$). In contrast, we find that childfree individuals ($M = 3.673$) and single parents ($M = 3.378$) are least satisfied with their neighborhood. Comparing these means, we find that childfree individuals are significantly less satisfied than both co-parents and not-yet-parents, and that single parents are significantly less satisfied than everyone except the childfree.

## 5 Discussion

Past research has consistently found that being a parent, or living in a household with children, is not associated with neighborhood satisfaction [2–19]. This lack of association is surprising because having children or being a parent is a significant life choice. To explore what may be happening, we used data from a representative sample of Michigan adults to compared the

neighborhood satisfaction of six reproductive status groups: co-parents, single parents, empty nesters, not-yet-parents, childless individuals, and childfree individuals. Our first research question asked *is reproductive status associated with neighborhood satisfaction*? Whether controlling for individual characteristics or not, we found that reproductive status explains a significant amount of variation in neighborhood satisfaction. Our second research question asked *how satisfied is each reproductive status group with their neighborhood, and which groups differ in their neighborhood satisfaction*? Through a series of group mean comparisons, we found that co-parents and not-yet-parents are the most satisfied, and are statistically significantly more satisfied with their neighborhoods than either childfree individuals or single parents.

## 5.1 A tale of two neighborhood experiences?

There is substantial variation across neighborhoods, and likewise substantial variation in residents' satisfaction with their neighborhoods. However, the mean levels of neighborhood satisfaction we report in Table 4 suggest that despite these variations, when it comes to reproductive status, there are two different ways that individuals experience their neighborhoods. One group—co-parents, not-yet-parents, childless individuals, and empty nesters—all have high average levels of neighborhood satisfaction. . .they are happy with where they live. A second group—childfree individuals and single parents—have significantly lower average levels of neighborhood satisfaction. . .they are less happy with where they live.

Because even small effects and differences may be statistically significant in a large $N = 946$ sample such as this, it is important to consider the practical significance of estimated effect sizes also. That is, although we find that childfree individuals and single parents are *statistically* less satisfied, are they *practically* less satisfied. Comparing the magnitudes of coefficients reported in Table 3 provides insight here. Relative to highly-satisfied co-parents, being a single parent is associated with having .7 fewer units of satisfaction on average. This is comparable to the difference in neighborhood satisfaction between a person who is maximally satisfied with life (SWLS = 7) and a person with a below-average satisfaction with life (SWLS = 4). It is also comparable to the difference in neighborhood satisfaction between a person for whom COVID made them much more stressed (COVID = 5) and a person for whom COVID made them much less stressed (COVID = 1). That is, the amount of lost neighborhood satisfaction experienced by single parents relative to co-parents is dramatic, paralleling the effects of a loss in satisfaction with life or of a pandemic.

## 5.2 Possible explanations

Because these data are cross-sectional and contain limited information about the respondents' neighborhoods, they do not allow us to draw conclusions about *why* single parents or childfree individuals experience less neighborhood satisfaction. One plausible explanation for single-parents' lower neighborhood satisfaction is their reduced disposable income, which limits their residential mobility [38]. Specifically, single parents may have less ability to choose where they live, and therefore are more likely to remain in neighborhoods that are not satisfying.

In contrast, because there is very little research on childfree individuals, it is less clear why childfree individuals experience less neighborhood satisfaction. Because we observe that childfree individuals experience substantially less neighborhood satisfaction and comprise over one-quarter of the adult population, more research on this group is needed. Although speculative, we briefly offer two possible explanations for this group's lower neighborhood satisfaction.

One possibility is that neighborhood satisfaction influences individuals' decision not to have children, such that living in an unsatisfying neighborhood may lead a person to choose not to have children (e.g., "the local schools are bad. . .I wouldn't want to raise kids here"). However, while this environment-shapes-behavior process could occur, it seems just as plausible that living in a highly satisfying neighborhood could lead a person to choose not to have children (e.g., "I love living in Manhattan, and couldn't stay here if I had kids"). Additionally, such explanations would more closely describe the decision making process not of child*free* individuals, but of child*less* individuals who wanted children but chose not to have them for personal, environmental, or biological reasons.

A second possibility is that individuals' decision not to have children influences their neighborhood satisfaction, such that those choosing not to have children are a poor fit or feel out-of-place in the most desirable neighborhoods [41]. In the United States, the most desirable neighborhoods are often located in suburbs, where the focus is on family-friendliness [36, 39] through the provision of child and parent-focused amenities such as schools, playgrounds, and youth activities. Childfree individuals living in such neighborhoods, which might yield high levels of satisfaction for many residents, may remain unsatisfied, while childfree individuals living in more urban or adult-focused settings may also encounter other factors (e.g., noise, congestion) that reduce their satisfaction.

## 5.3 Limitations

This study had several strengths, including a measurement strategy that allowed us to examine heterogeneity across six reproductive status groups and a large, representative sample. However, there are some limitations that should be considered when interpreting the results, and that highlight directions for future research. First, although our sample was representative, it was drawn from one state (Michigan) in one country (United States). More work is needed to determine whether our findings generalize to other locations. Second, the data we used are cross-sectional, and therefore do not allow us to draw causal conclusions, or to investigate the relationship between neighborhood satisfaction and changes in reproductive status across the life course. Future research could track, or retrospectively collect, the neighborhood satisfaction and reproductive trajectories of individuals over the life course [54]. Third, although we are able to control for demographic characteristics and well-being, because these data do not contain information about the neighborhoods themselves, we are unable to estimate the relative impact of reproductive status versus neighborhood characteristics on neighborhood satisfaction, or to consider potential interactions with neighborhood type. A recent meta-analysis has shown that neighborhood characteristics play a limited role in neighborhood satisfaction [20]. However, future studies may compare the role of reproductive status with perceived neighborhood characteristics such as crime or social cohesion, and may explore whether reproductive status plays the same role in different types of neighborhoods (e.g. high- vs. low-income neighborhoods; [55, 56]). Fourth, these data do not allow us to control for some demographic characteristics that are often considered in studies of neighborhood satisfaction, such as income, housing type, or tenure. Future studies should examine whether our findings replicate when these other potential influences on neighborhood satisfaction are controlled. Finally, while this study provides descriptive findings about *which* types of parents and non-parents are less satisfied with their neighborhoods, we are unable to determine *why* certain types of parents and non-parents are less satisfied. Future research is needed to unpack the mechanisms that link reproductive status and neighborhood satisfaction.

## 5.4 Conclusion

Despite these limitations, this study is among the first to explicitly measure the neighborhood satisfaction of individuals with distinct reproductive statuses in a representative sample, and uncovers a previously hidden heterogeneity. By distinguishing individuals with unique reproductive statuses, we have also clarified why prior work has found that having children in the household has no effect on neighborhood satisfaction. Specifically, we have demonstrated that parents and non-parents are not monolithic groups with respect to neighborhood satisfaction. Some types of parents and non-parents are quite satisfied with their neighborhoods while others are not. Moreover, the differences in neighborhood satisfaction between these contrasting groups were large in magnitude, and the groups experiencing the lowest average neighborhood satisfaction comprise one-third of Michigan's adult population.

Methodologically, this finding suggests that future research on neighborhood satisfaction should adopt more nuanced perspectives on reproductive status than a parent/non-parent dichotomy. Practically, this finding suggests that efforts to improve residents' neighborhood satisfaction must look beyond whether or not an individual is a parent. Although some current parents (i.e., co-parents) are quite satisfied with their neighborhoods, others (i.e., single parents) are not. Likewise, although some non-parents (i.e., not-yet-parents) are quite satisfied with their neighborhoods, others (i.e., childfree individuals) are not. Therefore, future neighborhood planning efforts should aim to understand the needs and interests of these groups and should explore opportunities to develop neighborhoods that are satisfying across reproductive statuses.

## Author Contributions

**Conceptualization:** Zachary P. Neal, Jennifer Watling Neal.

**Data curation:** Zachary P. Neal, Jennifer Watling Neal.

**Formal analysis:** Zachary P. Neal, Jennifer Watling Neal.

**Investigation:** Zachary P. Neal, Jennifer Watling Neal.

**Methodology:** Zachary P. Neal, Jennifer Watling Neal.

**Writing – original draft:** Zachary P. Neal, Jennifer Watling Neal.

**Writing – review & editing:** Zachary P. Neal, Jennifer Watling Neal.

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
