## [Decision Letter · Decision Letter 0]

8 Jul 2022

PONE-D-22-11374Neighborhood satisfaction and reproductive statusPLOS ONE

Dear Dr. Neal,

Thank you for submitting your manuscript to PLOS ONE. After careful consideration, we feel that it has merit but does not fully meet PLOS ONE’s publication criteria as it currently stands. Therefore, we invite you to submit a revised version of the manuscript that addresses the points raised during the review process.

We look forward to receiving your revised manuscript.

Kind regards,

Andrew T. Carswell

Academic Editor

PLOS ONE

Journal Requirements:

Reviewers' comments:

Reviewer's Responses to Questions

**Comments to the Author**

1. Is the manuscript technically sound, and do the data support the conclusions?

Reviewer #1: Yes

Reviewer #2: Yes

2. Has the statistical analysis been performed appropriately and rigorously? 

Reviewer #1: Yes

Reviewer #2: No

3. Have the authors made all data underlying the findings in their manuscript fully available?

Reviewer #1: Yes

Reviewer #2: Yes

4. Is the manuscript presented in an intelligible fashion and written in standard English?

Reviewer #1: Yes

Reviewer #2: Yes

5. Review Comments to the Author

Reviewer #1: This manuscript uses survey data to focus on the question of whether neighborhood satisfaction differs among persons based on their status as a household with children. It uses multiple categories to capture this construct in novel ways, and demonstrates heterogeneity across these groups in neighborhood satisfaction.

This is a well-done study, and I only have a small number of comments.

For the SWLS measure, it would be good to list all the questions that are part of this scale.

You do have a relatively small amount of missing data. Nonetheless, it would likely be better to use multiple imputation to address missingness, rather than listwise deletion. It likely will not change the results much, but would be more appropriate.

I appreciated Table 4, and showing the statistical significance between different categories.

In section 5.2 you hypothesize that SES may be a possible explanation. It appears that this is not included in the model? It would be reasonable to include this to assess this possible explanation.

Reviewer #2: This paper focuses on parental status as having a significant effect on neighborhood satisfaction. The authors find that single parents and childfree individuals experience significantly less neighborhood satisfaction than other groups. The paper explores the nuances of understanding neighborhood satisfaction and adds an often missing area of extant literature. For further consideration, I would like the authors to address my minor comments and critiques below:

Reproductive Statuses – I agree with the sentiments and framing of this identification or state of being but maybe considering another term like “parental statuses” (as noted in the paper) or “familial statuses” as ‘reproductive statuses’ embodies some assumptions around actual reproductive abilities. That is, categories such as “not-yet-parents” infers that people have the ability to be parents in the future.

Data – Which version of the ACS are the authors using in their analysis? It says 2016 ACS but is that the one, three, or five year estimates? Moreover, are these households clustered in any particular region or spatially distributed across the state, in urban, suburban and more rural spaces?

Timing of Data Collection – Were there any other sociopolitical events worth noting during the data collection time in Michigan? The governor kidnapping plot comes into mind but that was after the data collection period.

Covariates – The authors make several methodological choices which need further clarification. For example, for racial categories, the study includes White and Persons of Color. Given that this is a representative sample with almost 77% of it being White households, are diverse are racial and ethnic categories among households of color? Moreover, given that the percentage of White households is much higher than the state of Michigan (approx. 62% in 2021), is there any potential biases introduced into the analysis?

Moreover, income was not controlled for. Is this sample reflective of all households or particular income brackets? Other studies tend to focus on particular types of neighborhoods (such as low-income neighborhoods (See Ciorici and Dantzler 2019 or Jones and Dantzler 2021) or in localities undergoing significant redevelopment or relocation efforts (See Oakley et al. 2013). Given the framing of the paper, particularly the disaggregation of parental statuses, it seems that other coding choices should follow suite. Yet, the cross-sectional nature of the data may prohibit further disaggregation.

And lastly, what about housing tenure statuses? Do you have data on whether households are renting versus owning? In addition, is there data on their specific type of housing (i.e., apartment versus single family home)? Seems like these are important measures to consider given the diversity of neighborhoods surveyed in this study. Should none of these variables be available, the authors may want to add a couple sentences in their limitation section to address this.

Analyses – When including the covariates, the authors note that they may be association with reproductive statuses and neighborhood satisfaction. To control for the possible endogeneity here, have the authors considered other methodological approaches such as employing an SEM or HLM approach? While the analyses here is straightforward, I worry that some of the possible endogeneity imputed from this approach may be driving some of these estimates in particular ways.

Overall, I found the paper theoretically interesting and methodologically sound. However, I am offering a Revision for the purposes of addressing my earlier comments and critiques. I look forward to seeing a revised version of the paper.

6. PLOS authors have the option to publish the peer review history of their article (what does this mean?). If published, this will include your full peer review and any attached files.

Reviewer #1: No

Reviewer #2: No

---

## [Decision Letter · Decision Letter 1]

3 Aug 2022

Neighborhood satisfaction and reproductive status

PONE-D-22-11374R1

Dear Dr. Neal:

We’re pleased to inform you that your manuscript has been judged scientifically suitable for publication and will be formally accepted for publication once it meets all outstanding technical requirements.

Kind regards,

Andrew T. Carswell

Academic Editor

PLOS ONE

Additional Editor Comments (optional):

Reviewers' comments:

Reviewer's Responses to Questions

**Comments to the Author**

1. If the authors have adequately addressed your comments raised in a previous round of review and you feel that this manuscript is now acceptable for publication, you may indicate that here to bypass the “Comments to the Author” section, enter your conflict of interest statement in the “Confidential to Editor” section, and submit your "Accept" recommendation.

Reviewer #1: All comments have been addressed

Reviewer #2: All comments have been addressed

2. Is the manuscript technically sound, and do the data support the conclusions?

Reviewer #1: (No Response)

Reviewer #2: Yes

3. Has the statistical analysis been performed appropriately and rigorously? 

Reviewer #1: (No Response)

Reviewer #2: Yes

4. Have the authors made all data underlying the findings in their manuscript fully available?

Reviewer #1: (No Response)

Reviewer #2: Yes

5. Is the manuscript presented in an intelligible fashion and written in standard English?

Reviewer #1: (No Response)

Reviewer #2: Yes

6. Review Comments to the Author

Reviewer #1: (No Response)

Reviewer #2: I would like to thank the authors for their responses and revisions. I am recommending an Acceptance based off of the latest draft of the paper. I look forward to seeing this paper in print.

7. PLOS authors have the option to publish the peer review history of their article (what does this mean?). If published, this will include your full peer review and any attached files.

Reviewer #1: No

Reviewer #2: No

---

## [Editor Report · Acceptance letter]

5 Aug 2022

PONE-D-22-11374R1 

Neighborhood satisfaction and reproductive status 

Dear Dr. Neal:

I'm pleased to inform you that your manuscript has been deemed suitable for publication in PLOS ONE. Congratulations! Your manuscript is now with our production department. 

Kind regards, 

on behalf of

Dr. Andrew T. Carswell 

Academic Editor

PLOS ONE